# Neural-Guided Enumerative SAT Framework for Cryptographic Key Recovery

## Abstract

Boolean satisfiability (SAT) provides a principled framework for cryptanalysis by encoding cryptographic problems into conjunctive or algebraic normal forms. While the well-known (conflict-driven clause learning) CDCL-based solvers excel at handling SAT instances, their performance on cryptographic instances deteriorates quickly with increasing cipher complexity due to exponential search spaces. Recent neural approaches, ranging from end-to-end prediction to solver-integrated heuristics, show some promise yet suffer from scalability issues, limited recovery accuracy, or high computational overhead. In this work, we propose two complementary neural-guided enumerative SAT frameworks tailored for cryptographic key recovery. The first integrates lightweight neural predictions with traditional solvers: it predicts a subset of $k$ key variables as critical variables and enumerates the candidate assignments of these predicted variables, yielding up to a 5× speedup on the public benchmark SAT4CryptoBench. The second employs a discriminator trained to distinguish correct versus incorrect assignments in ANF encodings, enabling accurate pruning and propagation. On Simon datasets, this method achieves 79.5% full-key recovery accuracy, significantly surpassing prior neural approaches. Together, these frameworks bridge the gap between machine learning and classical SAT solving, offering scalable and efficient cryptanalysis.

## 1 Introduction

Boolean satisfiability (SAT) has long served as a cornerstone for tackling computationally hard problems in verification, planning, and cryptanalysis. In the context of cryptography, SAT solvers offer a principled approach to encode cryptographic problems into conjunctive normal form (CNF) or algebraic normal form (ANF), enabling automated key recovery and structural analysis. Traditional SAT solvers (Fleury & Heisinger, 2020; Soos, 2016), including CDCL-based methods and their variants, demonstrate strong performance on cipher-derived formulas. However, as cipher complexity increases, the resulting exponential search space renders these solvers prohibitively expensive.

Recent research has explored end-to-end neural network approaches for cryptanalysis, aiming to directly predict key bits or infer plaintext-ciphertext relationships from formal representations. For instance, NeuroSAT (Selsam et al., 2019) and CryptoANFNet (Zheng et al., 2024) demonstrate that neural networks can distinguish correct key bit assignments from incorrect ones with moderate accuracy when trained on CNF or ANF encodings. Similarly, Gohr (2019) shows that networks can effectively discriminate correct from incorrect differential patterns in block ciphers. Despite these promising results, such methods generally do not scale to full key recovery tasks and are often limited to serving as auxiliary tools for subsequent key search procedures. At the level of key bits recovery, their recovery accuracy remains limited, resulting in a substantial recovery accuracy performance gap compared to traditional SAT-based solvers in practical cryptanalysis scenarios.

In parallel, recent research has investigated neural-assisted SAT solvers, where machine learning models are integrated to guide or accelerate classical solving procedures. Common strategies include learning-based branching heuristics (Nejati et al., 2020; Nejati & Ganesh, 2020), variable selection mechanisms (Duan et al., 2020), and hyperparameter optimization (Dreczkowski et al., 2023). In the broader SAT domain, these techniques allow solvers to prioritize promising regions of the search space or to prune large portions of infeasible assignments, thereby improving efficiency. However, when applied to cryptographic SAT instances, such methods encounter several challenges. Cipher-derived formulas typically involve a large number of intermediate variables, making it difficult for

learned models to capture effective decision or branching strategies. Moreover, embedding neural networks into solver heuristics often incurs significant computational overhead, limiting their ability to scale efficiently to larger and more complex cryptographic SAT problems.

In this work, we address these challenges from two complementary neural-guided enumeration perspectives. First, we introduce a **heuristic solver-centered enumerative SAT framework** that integrates neural guidance with traditional SAT solving. Specifically, we train a neural network on plaintext–ciphertext pairs to predict a subset of $k$ key variables as critical variables. During solving, we enumerate the candidate assignments of these predicted variables, and the resulting CNF instances are submitted to a heuristic SAT solver. This framework directs the solver to focus on selected key variables while invoking the network only once at the start, substantially reducing computational overhead. In the experiments, our method achieves up to a $5\times$ speedup over the corresponding baseline solver on the recently published SAT4CryptoBench (Zheng et al., 2025) benchmark.

Second, we propose a **neural distinct-based enumerative SAT framework**, which generates candidate assignments over a small subset of key bits and trains a network to distinguish ANF instances corresponding to correct versus incorrect assignments. During solving, multiple derived ANF instances are produced by varying the selected subset of $k$ variables and their assignments according to the cipher's round-function clauses. We select the $k$-bit assignment classified by the network as correct for these $k$ variables. Then all key variables are subsequently recovered by iteratively propagating the known assignments and simply extracting assignments through the unit clause, which represents the simplest equation (e.g., $x_1 = 0$). This approach achieves 79.5% accuracy on Simon-6-32-64, enabling effective **full-key recovery** and overperforming the prior work (Zheng et al., 2024), which is nearly incapable of recovering the full key. **The highlights of this paper include:**

- **Heuristic Solver-Centered Framework:** We introduce a framework that integrates neural predictions with classical SAT solving. By focusing on a small subset of predicted key variables, it reduces computational overhead and achieves up to 5× speedup on SAT4CryptoBench.
- **Neural Distinct-Based Framework:** We introduce a framework that uses neural discrimination over candidate assignments of key-bit subsets to guide SAT solving. This approach enables effective full-key recovery and surpasses prior methods that fail to recover the full key.
- **Generality& Flexibility:** Our frameworks demonstrate strong generality and applicability to diverse ciphers and key sizes. These frameworks provide a unified, neural-guided SAT-based approach for cryptanalysis, offering a practical and innovative tool for full-key recovery.

## 2 RELATED WORKS AND PRELIMINARIES

**Differential-Cryptanalysis with Neural Networks.** Differential cryptanalysis, as an important technique in cryptanalysis, exploits the patterns of differences in plaintext pairs to infer information about the secret key. Gohr (2019) pioneered deep learning for differential cryptanalysis, showing that convolutional networks can distinguish correct from incorrect differential characteristics on reduced-round Speck32/64. This framework of differential analysis inspired follow-up works. Hou et al. (2021) extended attacks to Simon32/64 and other lightweight ciphers, while Benamira et al. (2021) systematically evaluated their strengths and limitations. Subsequent works explored reinforcement learning for automated differential trail discovery Bao et al. (2022), and improved training strategies So (2020); Seok & Lee (2024). Recent studies introduced mixture-based or enhanced neural differential attacks (Bao et al., 2023; Wu et al., 2024), and Singh et al. (2024) comprehensively surveys these developments. However, despite these advances, enumeration remains costly: networks mainly filter promising differential trails or key subspaces, but full key recovery still requires exhaustive enumeration over candidate keys with time complexity near $2^{31}$. These methods also struggle with single plaintext–ciphertext scenarios and require large training datasets (over $10^7$ instances). Besides, predictive accuracy at the key-recovery level remains limited, leaving a substantial gap compared to traditional SAT-based solvers.

**Neural SAT-based Cryptanalysis.** Algebraic cryptanalysis can be formulated as a satisfiability problem, motivating the application of neural networks to SAT solving for cryptography. Early approaches employed recursive neural networks to process CNF formulas as sequences, which later evolved into graph neural networks (GNNs) capable of directly capturing the structural information of formulas (Bünz & Lamm, 2017). NeuroSAT (Selsam et al., 2019) exemplifies this line of work, treating SAT solving as a prediction problem and using a neural network to predict satisfia-

bility on CNF instances from single-bit supervision, thereby providing an end-to-end framework for SAT solving. Related efforts further integrated the solving process with neural models, including GNNs (Cameron et al., 2020; Yan et al., 2023) and transformer-based architectures (Shi et al., 2023), to tackle boolean constraints arising in SAT solving. Furthermore, Li et al. (2024) proposed a dedicated benchmark to facilitate benchmarking and evaluation of GNN-based SAT solvers. Building on these developments, CryptoANFNet Zheng et al. (2024) extended the approach to cryptographic applications by training classifiers on ANF representations to distinguish correct from incorrect one-bit key assignments for key solving. However, these methods primarily focus on satisfiability with limited accuracy in key assignments, resulting in modest key-recovery performance and constrained practical applicability compared to traditional solvers.

**ML-based heuristic SAT Solvers.** A complementary line of research augments classical SAT solvers with machine learning guidance, including neural variable heuristics (Li & Si, 2022), clause and restart policy learning (Liang et al., 2018), portfolio selection (Zhang & Zhang, 2021), and hyperparameter optimization Dreczkowski et al. (2023); Wang et al. (2023). In the general SAT domain, these methods help prune large regions of the search space or prioritize promising assignments, effectively serving as a form of guided enumeration. For example, neural heuristics can bias branching toward likely-satisfiable subspaces (Jaszczur et al., 2020; Kurin et al., 2020), and learned clause deletion (Vaezipoor et al., 2020) or glue-variable prediction can reduce redundant exploration (Han, 2020). However, when applied to cryptographic SAT instances, these approaches face notable challenges. SAT4CryptoBench (Zheng et al., 2025) demonstrated that the integration of neural models frequently introduces significant computational overhead, limiting scalability and hindering generalization across ciphers and instance sizes. Moreover, cipher-derived SAT instances often involve many intermediate variables, making it difficult for learning heuristics.

To tackle these challenges, we propose two complementary frameworks. The heuristic solver-centered framework integrates neural-guided enumeration with classical strategies, reducing search space and solving time while preserving generalization. The neural distinct-based framework improves full key recovery by generating distinguishable derived instances and combining neural distinction with unit propagation, achieving higher accuracy on complex cryptographic SAT instances.

**CNF & ANF representation.** Boolean functions are often represented in two standard forms: conjunctive normal form (CNF) and algebraic normal form (ANF). CNF expresses a Boolean formula as a conjunction of clauses, each being a disjunction of literals, which is the standard input format for most SAT solvers. For example, the CNF formula $(x_1 \vee \neg x_2) \wedge (x_2 \vee x_3 \vee \neg x_4)$ contains the clause $(x_1 \vee \neg x_2)$, which in a CNF file is written as `1 -2 0`, and $x_1, \neg x_2$ are literals. ANF, on the other hand, represents Boolean functions as a sum of monomials over $\mathbb{F}_2$, where addition corresponds to XOR and multiplication corresponds to AND. For instance, $x_1 x_2 + x_3 + 1 = 0; x_1 + x_2 = 0$ is a ANF formula, where $x_1 x_2 + x_3 + 1 = 0$ is called a clause and $x_1, x_2, x_3$ are called literals.

**Cryptographic round function.** A round function is the basic building block of many symmetric-key cryptographic algorithms, such as block ciphers. Each round typically consists of nonlinear operations (e.g., S-boxes), linear diffusion layers (e.g., bitwise rotations, permutations, or matrix multiplications), and key addition steps. The iterative application of the round function ensures both confusion and diffusion, which are essential for cryptographic security. For SAT-based modeling, the round function is typically encoded as a set of symmetric clauses, and we refer to the clauses corresponding to the $i$-th encryption round as the $i$-th round-function clauses.

## 3 HEURISTIC SOLVER-CENTERED ENUMERATIVE SAT FRAMEWORK

### 3.1 BACKGROUND & OVERVIEW

Traditional heuristic-based SAT solvers, although highly efficient on general SAT benchmarks, often face considerable difficulties when applied to cryptographic SAT instances. As mentioned in SAT4CryptoBench (Zheng et al., 2025), these difficulties mainly arise from two sources: *i*) The search space induced by round-reduced block ciphers exhibits strong symmetries, which undermine the effectiveness of standard variable activity heuristics; *ii*) Solver branching decisions tend to oscillate on variables with limited informational value, such as intermediate variables in cipher round functions, leading to exponential growth in the search process. To address these issues, we propose the heuristic solver-centered enumerative SAT framework, which introduces a *neural-guided enumeration mechanism* designed to provide more effective guidance for the solver.

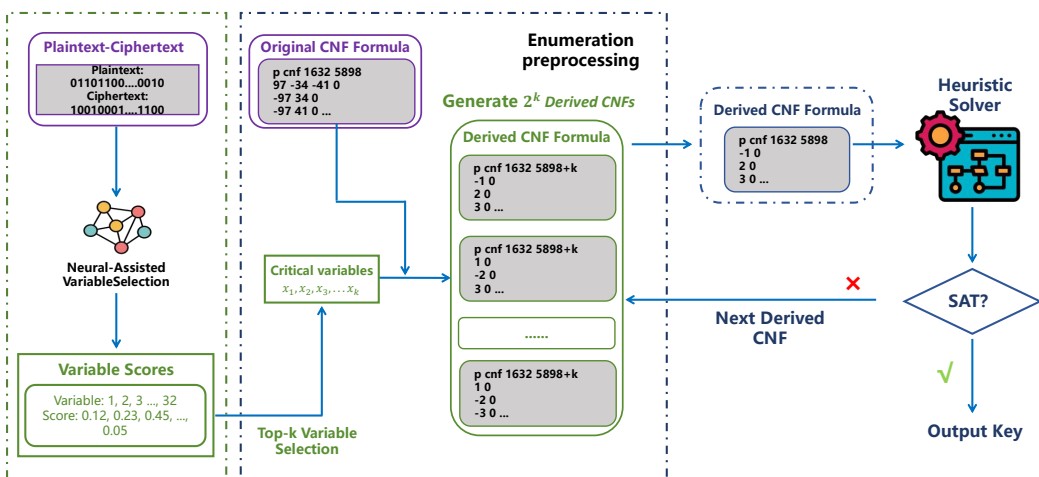

Figure 1: Overview of Heuristic solver-centered enumerative SAT framework. The framework first applies a neural network to score all key variables and selects the top-$k$ critical ones. Enumeration preprocessing then expands the original SAT instance into $2^k$ derived instances by fixing these variables, and a heuristic SAT solver sequentially solves the derived instances. Once a satisfiable instance is found, the fixed assignments together with the solver's output yield the full secret key.

**Overview.** The whole framework consists of three components: enumeration preprocessing, neural-assisted variable selection, and heuristic solver. As shown in Figure 1, we first select $k$ critical variables closely related to cryptographic problem solving (typically key variables). In enumeration preprocessing, the original CNF is expanded into $2^k$ derived instances by exhaustively enumerating these variables, and the heuristic solver is applied sequentially to solve these CNF instances until a satisfiable solution is found. Furthermore, to avoid manual selection of critical variables, a neural-assisted variable selection module scores variables based on plaintext–ciphertext pairs, and the top-$k$ are chosen as critical variables, providing more effective guidance for the solver.

**Remark.** In practice, the heuristic solver component can be replaced with different SAT solvers as the technology evolves. Thus, in this section, we focus primarily on the other two components.

## 3.2   ENUMERATION PREPROCESSING

Instead of allowing the solver to branch on uninformative intermediate variables repeatedly, our approach first fixes a subset of meaningful variables before the search begins. This idea is realized through an **enumeration preprocessing** step, where a small group of $k$ critical variables— chosen from the key variables of the cipher—are explicitly assigned in advance. By expanding the original SAT instance $F$ into multiple derived instances, each of which reflects one possible assignment of these $k$ variables, the solver is directed to explore only the reduced instances.

Formally, let $F$ denote the original SAT formula derived from the cryptographic problem, and let $V = \{x_1, x_2, \ldots, x_k\}$ be the set of selected $k$ critical variables (typically key variables in the encryption algorithm). For each assignment $\alpha = [\alpha_1, ..., \alpha_k] \in \{0, 1\}^k$, we construct a derived SAT instance $F_\alpha$ by augmenting $F$ with $k$ additional unit clauses $\{C_{x_i=\alpha_i}\}$: $F_\alpha = F \ \wedge \ \bigwedge_{i=1}^{k} C_{x_i=\alpha_i}$.

In practice, each unit clause $C_{x_i=\alpha_i}$ simply fixes one variable to either 0 or 1, e.g., $(x_i)$ or $(\neg x_i)$ in CNF and $(x_i)$ or $(x_i + 1)$ in ANF representation. This results in $2^k$ derived SAT instances $\{F_\alpha\}$, which are passed sequentially to the heuristic solver. Since each instance already has the $k$ variables determined, the solver avoids wasting search effort oscillating on these low-informative variables. In other words, equivalently, the solving process on the original instance $F$ is guided to branch on these strategically selected variables, thereby improving solving efficiency.

Importantly, this method is solver-agnostic, as the preprocessing only requires adding unit clauses and does not interfere with internal heuristics, thus making the method broadly applicable across different solvers. Besides, the parameter $k$ introduces a trade-off between the number of derived instances and the average solving time per derived instance, which will be evaluated in Section 5.2.

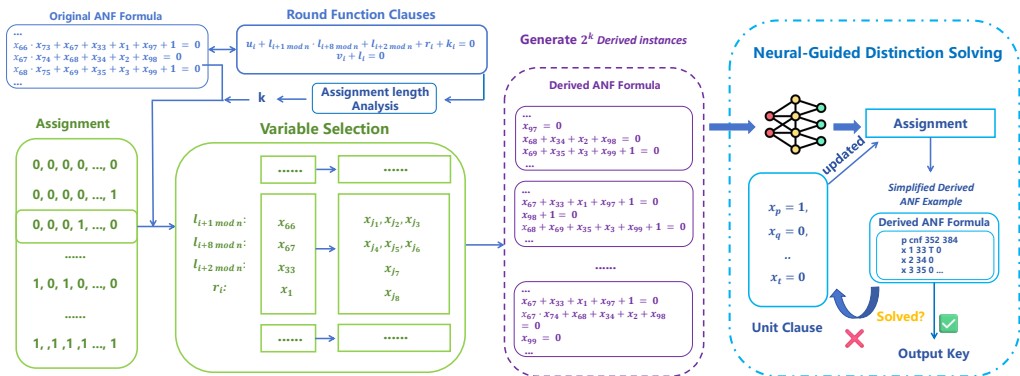

Figure 2: Overview of Neural distinct-based enumerative SAT framework. The framework first determines the assignment length $k$ by analyzing cipher-induced symmetric clauses. For each of the candidate assignments, the corresponding variable selection is derived from the round-function clauses, yielding $2^k$ candidate instances. A neural distinction model then separates correct from incorrect instances, and iterative unit propagation on the correct instance recovers the full secret key.

### 3.3 NEURAL-ASSISTED VARIABLE SELECTION

Another challenge arises from the search space induced by round-reduced block ciphers, which exhibits strong symmetries that undermine standard variable activity heuristics. In cryptographic SAT instances, symmetries appear as similar clause blocks generated by the cipher's rounds. In traditional solvers, branching on the key variables of the cipher is crucial for breaking these symmetric clause structures. However, due to each instance's different plaintext–ciphertext pair, arbitrary key selection rarely breaks this structure consistently. Thus, to address this issue, we propose a *neural-assisted variable selection* module that selects the most informative key variables for enumeration.

Formally, let $n$ be the length of plaintext/ciphertext and the input to the neural network be a pair of plaintext–ciphertext bit strings $(P, C)$, where $P, C \in \{0, 1\}^n$. The network processes the concatenated bit string and outputs a probability score for each candidate variable $x_j$ in the SAT instance. In other words, the neural network implements a mapping: $f_\theta : \{0, 1\}^{2n} \rightarrow [0, 1]^n$, where $n$ is the total number of key variables in the cryptographic instance and $\theta$ denotes the trainable parameters.

Specifically, the network $f_\theta$ consists of three parts: (i) an initial convolutional encoder that maps the concatenated plaintext–ciphertext vector $(P, C)$ into a hidden representation, (ii) a stack of $L$ residual blocks that extract nonlinear features from the hidden representation, and (iii) a prediction head that outputs a score for each key variable. Then, the whole selection process can be written as

$$h = \mathcal{R}_L \circ \mathcal{R}_{L-1} \circ \cdots \circ \mathcal{R}_1 \big( IC(P, C) \big) \tag{1}$$

$$f_\theta(P, C) = \text{softmax}\big(\mathcal{H}(h)\big), \quad V = \text{Top-}k\big(f_\theta(P, C)\big) \tag{2}$$

where $IC(\cdot)$ is a convolutional block and $\mathcal{H}(\cdot)$ is an MLP. $\mathcal{R}_\ell(\cdot)$ denotes the $\ell$-th residual block applying a residual mapping with activation and normalization. Here, we select the top-$k$ key variables with the highest scores as the set $V$ of critical variables.

In practice, we take the $k$ key variables that lead to the fastest solving performance—identified through enumeration preprocessing over the derived instances—as labels to train the network (see Section 5.2 for details). By concentrating on the top-$k$ effective variables, the solver is guided toward a smaller yet more informative search space, where the inherent symmetries of cryptographic SAT instances are more effectively broken. This neural-guided method eliminates random variable selection and uses the underlying cryptographic structure to guide the enumeration preprocessing.

## 4 NEURAL DISTINCT-BASED ENUMERATIVE SAT FRAMEWORK

### 4.1 BACKGROUND & OVERVIEW

Existing neural approaches to cryptographic key recovery still face important challenges. Differential distinguishers require costly enumeration and large training datasets, while lacking generality for single plaintext–ciphertext scenarios. Likewise, neural SAT classifiers tend to focus on satisfiability prediction to distinguish correct from incorrect one-bit key assignments for key solving, but

their bit-level distinction accuracy remains modest for reliable key recovery. Faced with these challenges, we propose the neural distinct-based enumerative SAT framework, which enumerates $k$ key variables across different assignments and selection based on cipher-induced structural clauses and introduces a *neural-guided distinction mechanism* to enable full key recovery via unit propagation.

**Overview.** The whole framework consists of two parts: candidate derived instance generation and neural-guided distinction solving. As shown in Figure 2, we first determine the length of the enumerated assignment as the number of key variables required to fully simplify the relevant shallow round-function clauses. Then, we enumerate all $2^k$ assignments, substituting each into the original instance and selecting the key variables that fully simplify the relative round-function clauses as the variable selection corresponding to the assignment. This yields $2^k$ derived instances. Then we use a neural network to distinguish the correct instance and recover all the variables in this instance by propagating known values and extracting assignments via unit clauses (e.g., $x_1 = 0$).

**Remark.** In practice, since ANF provides a more concise representation than CNF, we adopt ANF as the SAT representation. Thus, in this section, SAT instances are assumed to be in ANF form.

## 4.2 CANDIDATE DERIVED INSTANCE GENERATION

The limited recovery accuracy of existing neural SAT approaches often stems from the minimal differences between derived instances produced by single-bit key assignments. For example, in the ANF representation, the distinction between two corresponding derived instances only lies in the constant term of a single clause, while the majority of clauses, which are structurally symmetric due to the cipher's round functions, remain identical. Therefore, we focus on breaking the symmetries of round-function clauses to generate candidate derived instances with greater distinction.

Here, we take the round-function clauses of the Simon dataset from SAT4CryptoBench (Zheng et al., 2025) as an example, where each round uses the same seed key. Let $L = \overline{l_0, l_1, \ldots, l_{n-1}}$ denote the left (high-order) input of the round function, $R = \overline{r_0, r_1, \ldots, r_{n-1}}$ the right (low-order) input, $K = \overline{k_0, k_1, \ldots, u_{n-1}}$ the round key, $U = \overline{u_0, u_1, \ldots, u_{n-1}}$ the left (high-order) output, and $V = \overline{v_0, v_1, \ldots, v_{n-1}}$ the right (low-order) output, where $n$ is the block size and $l_i, r_i, k_i, u_i, v_i \in \{0, 1\}$.

**Assignment length Analysis.** We first analyse the symmetric round-function clauses to determine the enumerated assignment length $k$. The round function of Simon can then be written as follows:

$$U = ((L \lll 1) \cdot (L \lll 8)) \oplus (L \lll 2) \oplus R \oplus K$$
$$V = L \tag{3}$$

The corresponding round-function clauses in the ANF are given as follows, for $i = 0, 1, \ldots, n-1$:

$$u_i + l_{i+1 \mod n} \cdot l_{i+8 \mod n} + l_{i+2 \mod n} + r_i + k_i = 0$$
$$v_i + l_i = 0 \tag{4}$$

Each round introduces $m$ structurally symmetric clauses. Since clauses in the middle rounds typically involve a larger number of key variables, we focus on the first three and the last three rounds. Guided by the following reasoning, we set $k = 8$ as the assignment length.

> **Analysis**
>
> In the first round, $V$ corresponds to the plaintext and is constant, while each bit in $U$ depends only on a single key bit. In the second round, every bit in $L$ is influenced by one key bit, while $R$ remains constant. Hence, knowing the values of just three key bits—$l_{i+1 \mod n}$, $l_{i+8 \mod n}$, and $k_i$—is sufficient to determine $u_i$. In the third round, the dependency becomes richer: each bit in $L$ involves 3 key variables, and $R$ is tied to one key bit. To fully simplify the $i$-th clause in this round, we must determine $l_{i+2 \mod n}$ and $k_i$, and further choose an assignment that forces either $l_{i+1 \mod n}$ or $l_{i+8 \mod n}$ to 0.
>
> For the last three rounds, the symmetry of Simon yields similar results. Specifically, once the $i$-th clause in the third round is fully simplified, the corresponding clause in the third-to-last round is also fully simplified. This analysis shows that at least 8 variables are required.

**Generation Process.** Based on the assignment length analysis, as shown in Figure 2, we generate the derived instances in three steps. First, we check the special case where $l_i = 1$ for all $i \in$

$1, 2, \ldots, n$ in the third-round clauses, which only requires verifying simple linear Boolean equations from the first two rounds clauses. If this condition holds, the solution can be obtained directly. Second, we generate all $2^k$ possible $k$-bit assignments. Third, for each specific assignment, we substitute the k-bit assignment into the $i$-th $(i = 1, \ldots, n)$ third round-function clauses to compute $l_{i+1 \bmod n}$, $l_{i+2 \bmod n}$, $r_i$, and $k_i$, and evaluate the resulting candidates. The instance with the greatest simplification is chosen as the derived instance for that assignment.

**Remark.** Notably, in the first step, we handle a special case to ensure that at least one of these derived instances is satisfiable, and this is proved in Appendix C. Besides, in the analysis, we rely only on the symmetric clause blocks denoted in Equation 4, without referring to the round functions in Equation 3. This implies that the assignment length can be determined directly from key variables and shallow symmetric clause blocks in the instance, without requiring knowledge of the underlying cryptographic scheme. Moreover, in practice, since the type of round function clauses is fixed, we only need a finite table to generate derived instances, making the overhead negligible.

### 4.3 NEURAL-GUIDED DISTINCTION SOLVING

After generating the $2^k$ derived SAT instances, we use a neural network to distinguish the correct instance. Formally, let $F$ be the original SAT instance in ANF and $\mathcal{A}$ be the assignment set of the derived SAT instances. For each assignment $a = [a_1, ..., a_k] \in \{0, 1\}^k$ with its related subset of key variables $V_a = \{x_{i_1}, ..., x_{i_k}\}$, the derived instance is constructed as: $F_a = F \ \wedge \ \bigwedge_{j=1}^{k} C_{x_{i_j} = a_j}$.

To distinguish these instances, we train a GNN-based neural network $\mathcal{N}_\theta$ to map an ANF instance to a confidence score $s_a \in [0, 1]$ indicating whether $a$ is likely to correspond to a correct partial key assignment: $s_a = \mathcal{N}_\theta(F_a), \quad a \in \mathcal{A}$. Specifically, the network $\mathcal{N}_\theta$ converts the SAT instance into an ANF graph proposed in the prior work (Zheng et al., 2024), iteratively refines the embeddings of literals and clauses, and outputs the classification logits through a linear classifier, as follows:

$$
\textbf{Iteration process:} \qquad h_l^{(t)} = \text{AGG}_l \left( \left\{ h_c^{(t-1)} | c \in \mathcal{N}(l) \right\}, h_l^{(t-1)} \right)
$$

$$
h_{c,\text{pos}}^{(t)}, h_{c,\text{neg}}^{(t)} = (\text{AGG}_{c,\text{pos}}, \text{AGG}_{c,\text{neg}}) \left( \left\{ h_l^{(t-1)} | l \in \mathcal{N}(c) \right\}, h_c^{(t-1)} \right) \qquad (5)
$$

$$
\textbf{Linear classifier:} \qquad s_a = C_{\text{vote}}(h_{c,\text{pos}}, h_{c,\text{neg}})
$$

where $\mathcal{N}()$ denotes the set of the neighbor nodes. $C_{\text{vote}}(\cdot)$ denotes an MLP, and $\text{AGG}(\cdot)$ represents the aggregation function. $h_{c,\text{pos}}^{(t)}, h_{c,\text{neg}}^{(t)}$ represent the embeddings of the $t$-th iteration for the positive clauses and the relevant negative clauses, respectively. $h_{c,\text{pos}}, h_{c,\text{neg}}$ represent the embeddings of the final iteration. Then, during solving, the network evaluates all derived instances $\{F_a\}$, and the assignment with the highest confidence score is selected: $a^* = \arg\max_{a \in \mathcal{A}} s_a$.

**Unit propagation.** Given the selected $k$-bit assignment $a^* = [a_1^*, \ldots, a_k^*]$, we recover all variables in the SAT instance through iterative unit propagation. Starting from the solution set $S = \{x_{i_1} = a_1^*, \ldots, x_{i_k} = a_k^*\}$, we substitute known values into clauses (e.g., $x_{i_1} + x_3 = 0$ becomes $x_3 + a_1^* = 0$) and extract new assignments from the resulting unit clauses, such as $x_3 + a_1^* = 0$, which contains only a single variable, thereby expanding $S$. This process iterates until all variables are determined or a conflict occurs. When no unit clause is available, we assign 0 to the first variable in the shortest clause to continue. Although this heuristic may slightly reduce recovery accuracy, experiments in Section 5.3 demonstrate that the method still achieves a high full-recovery success rate.

In practice, we use the aggregation functions of the ANF-based network in the prior work (Zheng et al., 2024), and the detailed structure is shown in Appendix B. Besides, we use the classification label to train this network and use the logits corresponding to satisfiable as the confidence scores.

## 5 EXPERIMENTS

In this section, we conducted experiments on the proposed two enumerative SAT frameworks to assess their effectiveness compared to the heuristic solvers and neural approaches. Experimental settings, including computation resources, are provided in Appendix A.

### 5.1 DATASETS

We use the Simon and Cipher datasets from SAT4Cryptobench (Zheng et al., 2025) for both training and testing. Specifically, Simon corresponds to SAT instances generated from the block cipher

Table 1: Comparison of average runtime (s/instance) across different methods. HSESAT: heuristic solver-centered enumerative SAT framework; Random: HSESAT using random variable selection; Base: original solvers.

| Solver | Methods | Simon 12-32-64 | Cipher 12 |
|---|---|---|---|
| **Kissat** | Base | 776.53 | 1028.96 |
| | Random | 376.08 | 414.99 |
| | HSESAT | **141.80** | **163.22** |
| **MapleSAT** | Base | 464.65 | 696.00 |
| | Random | 1462.59 | 993.81 |
| | HSESAT | **436.66** | **367.33** |
| **Crypto minisat** | Base | 2672.83 | 3240.66 |
| | Random | 3411.04 | 3550.42 |
| | HSESAT | **523.92** | **423.00** |

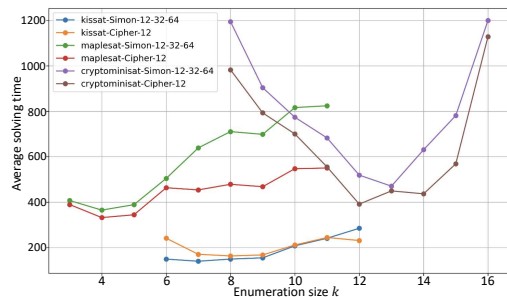

Figure 3: Enumeration Size $k$ vs. Average solving time (s/instance) in the heuristic solver-centered enumerative SAT framework for different solvers and datasets. To clearly illustrate the results, average solving times are plotted over tailored $k$ ranges for each solver.

Simon, a Feistel cipher that has been widely used in cryptographic research. In contrast, Cipher, designed to mimic ARX-based cipher decryption with unknown round functions, poses additional challenges due to its partially hidden structures. It evaluates solvers on ARX-dominant constraints without any known algebraic backdoor, thereby providing a "black-box" hardness that more closely resembles real-world ciphers. In the following experiments, Simon-$r$-$n$-$2n$ denotes the SAT dataset generated from the Simon cipher with $r$ encryption rounds, block size $n$, and plaintext/ciphertext length $2n$, and Cipher-$k$ refers to the SAT dataset constructed from the Cipher with $k$ rounds.

## 5.2 EVALUATION OF HEURISTIC SOLVER-CENTERED ENUMERATIVE SAT FRAMEWORK

**Baselines & Settings.** We evaluate the effectiveness of our heuristic solver-centered enumerative SAT framework. Here, we use several representative SAT solvers, including Kissat (Fleury & Heisinger, 2020), CryptoMiniSat (Soos, 2016), and MapleSAT (Nejati et al., 2021), as baselines for comparison. These solvers are widely used for large and complex SAT instances. We compare two settings: (i) the baseline performance of each solver on the original SAT instances, and (ii) the performance of the same solvers when applied within our framework, where $2^k$ derived instances are generated and key variables are selected using our neural network model before being passed to the solver. Besides, in our framework, we use the $k$ key variables, which yields the fastest solving performance as labels. Specifically, we measure the solving performance of the $k$ key variables using the average solving time of the derived instances obtained through enumeration preprocessing on Kissat. The labels are represented as $n$-dimensional $k$-hot vectors, with $k$ positions set to 1 and the rest set to 0, and the network is trained using cross-entropy loss.

**Evaluation of enumeration size $k$'s Trade-off**. We conduct experiments to analyze the effect of the enumeration size $k$ using a subset of the datasets. A larger $k$ generates more derived instances ($2^k$), increasing computational cost but enhancing simplification and solver efficiency, while a smaller $k$ reduces overhead at the expense of weaker symmetry breaking. To examine this trade-off, we vary $k$ and plot solver runtimes on different datasets within our framework. As shown in Figure 3, we find that across solvers and datasets, runtime first decreases and then increases as $k$ grows, indicating the existence of an optimal $k$, which strikes the practical balance between overhead and effectiveness.

**Evaluation of efficiency.** After analyzing the impact of the enumeration size $k$, we conduct experiments to evaluate the average solving time of each solver with the optimal $k$ on different datasets, as summarized in Table 1. Notably, for each dataset, we conduct experiments on a separate set of 200 samples, distinct from those used in the evaluation of $k$. The results demonstrate that our framework consistently reduces solving time compared with baseline solvers across different cryptographic SAT datasets. In particular, Kissat and Cryptominisat benefit significantly from our framework, achieving up to 5× speedup. Besides, ablation results in Table 1 reveal that replacing neural-assisted variable selection with random selection markedly degrades performance, highlighting its critical role in our framework. Moreover, we conduct additional experiments with the same solver, Kissat, under different optimization strategies. The results, detailed in Table 4 of Appendix D, show that our framework consistently provides substantial speedups across different optimization methods.

Table 2: Full key recovery accuracy performance comparison of different solvers. Here, NDESAT denotes our neural distinct-based enumerative SAT framework.

| Solver | Simon-3-16-32 | Simon-3-32-64 | Simon-6-16-32 | Simon-6-32-64 |
|---|---|---|---|---|
| **NDESAT(ours)** | 98.5% | 97.4% | 83.0% | 79.5% |
| **NeuroSAT** | 1.24% | 0.03% | <0.01% | <0.01% |
| **CryptoANFNet** | 2.37% | 0.18% | 0.24% | <0.01% |

**Remark.** Embedding heuristic solvers into our enumerative framework yields significant performance gains. Notably, the solver BASIN in SAT4CryptoBench (Zheng et al., 2025) can also use efficient bit-level enumeration of $2^n$ feasible keys (where $n$ is the key length) to achieve high efficiency. However, this approach is limited by key length and requires full cryptographic knowledge. In contrast, our framework works directly on SAT instances without such assumptions, and remains general and extensible: as heuristic solvers improve, it can further enhance their performance.

### 5.3 Evaluation of Neural distinct-based enumerative SAT framework

**Baselines & Settings.** We evaluate our neural distinct-based enumerative SAT framework in terms of both full key recovery accuracy and efficiency. Here, we compare against two representative neural SAT solvers from SAT4CryptoBench (Zheng et al., 2025): NeuroSAT (Selsam et al., 2019) and CryptoANFNet (Zheng et al., 2024), which are designed to directly reason over general SAT instances in CNF and cryptographic-specific ANF encodings, respectively. Besides, we use the binary cross-entropy loss to train the network in our framework, where we treat satisfiable derived instances from candidate generation as positive samples and randomly select an equal number of unsatisfiable instances as negative samples.

**Evaluation on Full Key Recovery** We conduct experiments on the task of full key recovery. For each cryptographic SAT instance, the baseline neural solvers are applied $n$ times, where $n$ is the key length. In each run, they evaluate two derived instances obtained by assigning either 0 or 1 to a specific key bit, and the bit value is determined by the higher prediction score. In contrast, our framework employs the neural network $\mathcal{N}_\theta$ to distinguish among the $2^k = 256$ derived instances, followed by unit propagation to recover the full solution. Table 2 presents the full key recovery accuracy results, which show that while previous methods are almost incapable of recovering the full key, our approach attains nearly 80% accuracy in full key recovery, highlighting its effectiveness. Furthermore, Table E in Appendix E presents a simple efficiency comparison. Overall, our method achieves a substantial accuracy improvement in 32-bit key recovery with only $2^3 \times$ enumeration.

**Remark.** Previous approaches achieve moderate accuracy in predicting satisfiability for plaintext–ciphertext pairs or single-bit assignments, but their performance collapses for 32-bit or longer keys due to exponential compounding. Our framework overcomes this by combining enumeration with unit propagation, avoiding repeated predictions and achieving higher accuracy. Despite the limitations compared to heuristic solvers, its advantage enables practical deployment of learning-based solvers, and we expect its applicability to further expand as GPU memory capacity increases.

## 6 Conclusion and Outlook

In this paper, we have presented two complementary neural-guided enumerative SAT frameworks for cryptographic key recovery. The heuristic solver-centered framework efficiently integrates neural predictions with traditional SAT solving, focusing on selected key variables and achieving up to a $5\times$ speedup. The neural distinct-based framework leverages neural discrimination over subsets of key bits to enable accurate iterative full-key recovery, attaining 79.5% accuracy on Simon-6-32-64 and significantly outperforming prior neural approaches. This demonstrates the practical effectiveness of the neural-guided frameworks for complex cryptographic SAT solving.

**Future Work.** Our frameworks are broadly applicable to diverse cipher structures, key sizes, and SAT encodings, highlighting their generality and flexibility. Future work could explore scaling to larger ciphers, employing more advanced neural architectures, or developing adaptive variable selection strategies. Integration with emerging SAT solver optimizations or GPU acceleration may further improve efficiency, underscoring the potential of neural-guided SAT methods as a general and scalable approach for cryptographic analysis and other hard computational problems.

ETHICS STATEMENT

This work adheres to the ICLR Code of Ethics. Our research focuses on SAT-based cryptanalysis in academic settings. The goal is to advance solver and machine learning methodology, not to provide deployable attack tools. No human subjects or sensitive information are involved, and all authors confirm compliance with research integrity and responsible dissemination.

REPRODUCIBILITY STATEMENT

All datasets used in our experiments are publicly available. Experiment settings are shown in the appendix, with additional proofs and model architectures provided. An anonymized link to a working example is available at: `https://anonymous.4open.science/r/Neural-Guided-Enumerative-SAT-Framework-F61B`. These resources collectively ensure that our results can be independently verified and extended by the research community.

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

## A  EXPERIMENT SETTING

### A.1  COMPUTATIONAL RESOURCES

All experiments were conducted on a dedicated high-performance computing cluster equipped with NVIDIA A100 GPUs. Training and inference for all neural components were performed entirely on these GPUs using the PyTorch framework, with CUDA and cuDNN enabled for efficient acceleration. CPU resources were used only for data loading and SAT-solving routines, while GPU devices were exclusively allocated to neural network training to avoid inter-process contention. The neural network itself is lightweight and computationally efficient, imposing only minimal demands on GPU resources and allowing fast and stable training.

### A.2  DETAIL OF DATASETS

In our experiments, we make use of the SATBENCHMARK datasets, with a particular focus on the Simon and Cipher series. All experimental evaluations are conducted on the official test sets from SATBENCHMARK, ensuring fair comparison and consistency with prior work. The average numbers of variables and clauses for the datasets we used are summarized in Table 3.

Table 3: Dataset size statistics used in our experiments.

| Dataset | Average #Variables | Average #Clauses |
|---|---|---|
| Simon-3-16-32-ANF | 32 | 48 |
| Simon-3-32-64-ANF | 48 | 72 |
| Simon-6-16-32-ANF | 80 | 96 |
| Simon-6-32-64-ANF | 120 | 144 |
| Cipher-12-CNF | 864 | 7424 |
| Simon-12-32-64-CNF | 1632 | 5888 |

Beyond evaluation, we further utilize the CNF encodings released in SATBENCHMARK to reconstruct the original cryptographic samples. In particular, we extract the plaintext–ciphertext pairs underlying the CNF formulas, which serve as supervised training data for the neural components of our framework.

## B  ANF-BASED MODEL ARCHITECTURE

This section provides the complete architectural details of the network we use in our neural distinct-based enumerative SAT framework, which is an ANF-based network in SAT4CryptoBench(Zheng et al., 2025), including the message-passing mechanism and the attention-based prediction head.

**Message-Passing Network**  We randomly initialize embeddings for vanilla literals and paired complementary clause nodes. The message-passing process consists of $T$ iterations, during which node embeddings are iteratively updated by aggregating information from their graph neighbors.

In each iteration, clause embeddings are first updated. A clause node receives messages from connected monomial nodes, which themselves aggregate messages from literal nodes, as well as from their complementary clause node. The update equations are as follows:

$$L_{l2c}^{(t)} = L_{l2l}(M_{l2l}^T L^{(t)})$$
$$[C_{m,\text{pos}}^{(t)}, C_{m,\text{neg}}^{(t)}] = M_{l2c}^T L_{\text{msg}}(L_{l2c}^{(t)})$$
$$(C_{\text{pos}}^{(t+1)}, C_{h,\text{pos}}^{(t+1)}) \leftarrow C_{u,\text{pos}}([C_{h,\text{pos}}^{(t)}, C_{\text{neg}}^{(t)}, C_{m,\text{pos}}^{(t)}]) \tag{6}$$
$$(C_{\text{neg}}^{(t+1)}, C_{h,\text{neg}}^{(t+1)}) \leftarrow C_{u,\text{neg}}([C_{h,\text{neg}}^{(t)}, C_{\text{pos}}^{(t)}, C_{m,\text{neg}}^{(t)}])$$

Here, $M_{l2l}$ and $M_{l2c}$ denote sparse adjacency matrices connecting literals to monomials and literals to clauses, respectively. The functions $L_{l2l}$ and $L_{\text{msg}}$ are multilayer perceptrons (MLPs) applied to literal messages. Clause updates are performed via layer-normalized LSTMs, $C_{u,\text{pos}}$ and $C_{u,\text{neg}}$.

Next, literal embeddings are updated by receiving messages from clauses, via intermediate monomial aggregation:

$$L_{c2l}^{(t)} = M_{l2c} C_{\text{msg}}([C_{\text{pos}}^{(t)}, C_{\text{neg}}^{(t)}]) \tag{7}$$

$$L_m^{(t)} = M_{l2l} L_{l2m}(L_{c2l}^{(t)}) \tag{8}$$

$$(L^{(t+1)}, L_h^{(t+1)}) \leftarrow L_u([L_h^{(t)}, L_m^{(t)}]) \tag{9}$$

where $C_{\text{msg}}$ and $L_{l2m}$ are MLPs, and $L_u$ is a layer-normalized LSTM that updates the literal state.

**Feedforward Prediction Head with Attention** After $T$ rounds of message passing, the refined clause embeddings are used to predict the satisfiability of the formula. Unlike Zheng et al. (2024), which aggregates clause embeddings by simple summation, we incorporate an attention mechanism inspired by Graph Attention Networks (GAT). Specifically, attention weights are computed across all positive and negative clause nodes to obtain a weighted aggregation:

$$\alpha_i = \frac{\exp(\text{LeakyReLU}(a^\top [W_c c_i \| W_c \bar{c}_i]))}{\sum_j \exp(\text{LeakyReLU}(a^\top [W_c c_j \| W_c \bar{c}_j]))} \tag{10}$$

$$s = \sigma\left(\sum_i \alpha_i \cdot (W_c c_i + W_c \bar{c}_i)\right) \tag{11}$$

where $c_i$ and $\bar{c}_i$ denote the $i$-th positive and negative clause embeddings, respectively, $W_c$ is a learnable projection matrix, $a$ is the attention vector, and $\| \cdot \|$ denotes vector concatenation. The final satisfiability score $s \in [0, 1]$ is obtained by applying the sigmoid activation $\sigma(\cdot)$.

The model is trained with binary cross-entropy loss between the predicted score $s$ and the ground truth satisfiability label $y \in \{0, 1\}$.

## C   PROOF OF THE COMPLETENESS FOR CANDIDATE DERIVED INSTANCE GENERATION

**Completeness of Generation process:** The generation process is shown as follows:

1. Check the special case where $l_i = 1$ for all $i \in 1, 2, \ldots, n$ in the third-round clauses, which only requires verifying simple linear Boolean equations from the first two rounds clauses. If this condition holds, the solution can be obtained directly.
2. Generate all $2^k$ possible $k$-bit assignments.
3. For each specific assignment, substitute the k-bit assignment into the $i$-th ($i = 1, \ldots, n$) third-round clauses to compute $l_{i+1 \bmod n}$, $l_{i+2 \bmod n}$, $r_i$, and $k_i$, and evaluate the resulting candidates. The instance with the greatest simplification is chosen as the derived instance for that assignment.

where the corresponding round-function clauses in the ANF are given as follows, for $i = 0, 1, \ldots, m-1$: $u_i + l_{i+1 \bmod n} \cdot l_{i+8 \bmod n} + l_{i+2 \bmod n} + r_i + k_i = 0$, $v_i + l_i = 0$.

*Proof.* The completeness of the generation process means that the correct derived instances are always contained within the generated $2^k$ instances. The completeness of the generation process implies that the correct derived instance is always included among the generated $2^k$ instances. Therefore, it suffices to prove that at least one satisfiable instance exists within the $2^k$ derived instances.

Based on the completeness of the generation process, we consider the following two cases:

1. **All third-round key variables satisfy $l_i = 1$ for $i \in \{1, 2, \ldots, n\}$.**
   In this scenario, the correct key is obtained directly in the first step.
2. **Not all third-round key variables satisfy $l_i = 1$ for $i \in \{1, 2, \ldots, n\}$.**
   In this case, there exists a $k$-bit assignment such that $l_{p+1 \bmod n} = 0$ for some $p \in \{1, 2, \ldots, n\}$. For the corresponding round-function clause containing $l_{p+1 \bmod n}$, substituting this $k$-bit assignment ensures that $l_{p+1 \bmod n}$, $l_{p+2 \bmod n}$, $r_p$, and $k_p$ are all known. Moreover, since $l_{p+1 \bmod n} = 0$, we have

$$l_{p+1 \bmod n} \cdot l_{p+8 \bmod n} = 0,$$

Table 4: Comparison of average runtime (s/instance) for solvers under different optimization strategies across frameworks. HSESAT: heuristic solver-centered enumerative SAT framework; Random: HSESAT using random selection instead of neural-assisted variable selection; Base: original solvers.

| Solver | Methods | Simon 12-32-64 | Cipher-12 |
|--------|---------|----------------|-----------|
| **Kissat** | Base | 776.53 | 1028.96 |
| | Random | 376.08 | 414.99 |
| | HSESAT | 141.80 | 163.22 |
| **Kissat-restart** | Base | 333.42 | 360.41 |
| | Random | 431.05 | 456.45 |
| | HSESAT | 177.17 | 215.40 |
| **Kissat-HO** | Base | 415.62 | 386.79 |
| | Random | 366.49 | 426.14 |
| | HSESAT | 144.50 | 163.53 |

so this clause is fully simplified under this assignment. (In practice, even when multiple clauses achieve the same level of simplification, a few steps of unit propagation are sufficient to identify satisfiable instances that the network can reliably distinguish.) Therefore, the instance corresponding to the true key variables must appear among the generated derived instances.

Combined with the above two cases, we have proved the completeness of the generation process. □

## D    EXPERIMENTS ON KISSAT UNDER DIFFERENT OPTIMIZATION STRATEGIES

We further test Kissat under different optimization strategies, including the adaptive restart policy (Li et al., 2022) and the hyperparameter optimization (Dreczkowski et al., 2023). As shown in Table 4, our framework consistently delivers substantial speedups across these strategies. These results highlight the robustness and generality of our framework: regardless of the specific optimization strategies embedded in solvers, the heuristic solver-centered enumerative SAT framework consistently provides notable performance gains, further underscoring the essential role of neural-assisted variable selection in accelerating SAT solving on cryptographic instances.

## E    RESULTS OF THE EFFICIENCY OF NEURAL DISTINCT-BASED ENUMERATIVE SAT FRAMEWORK

Table 5: Comparison of the efficiency across different solvers and datasets. (Average runtime: ms/instance) Here, NDESAT denotes our neural distinct-based enumerative SAT framework.

| Solver | Simon-3-16-32 | Simon-3-32-64 | Simon-6-16-32 | Simon-6-32-64 |
|--------|---------------|---------------|---------------|---------------|
| **NDESAT(ours)** | 297 | 322 | 336 | 375 |
| **NeuroSAT** | 18 | 33 | 41 | 52 |
| **CryptoANFNet** | 20 | 40 | 47 | 66 |

## F    LLM USAGE STATEMENT

Large Language Models (LLMs) were used solely for grammar correction and language polishing. No part of the research ideation, methodology, experimental design, analysis, or substantive writing was generated by LLMs. The authors take full responsibility for the content of this paper.

