# OpenReview forum: "Neural-Guided Enumerative SAT Framework for Cryptographic Key Recovery"
_ICLR.cc/2026/Conference — Submitted to ICLR 2026_

### Official Review · Reviewer_xeJ3 · 2025-10-20

**Soundness:** 3
**Presentation:** 3
**Contribution:** 3
**Rating:** 4
**Confidence:** 3

**Summary:**

This paper proposes two neural-guided enumerative SAT frameworks for cryptographic key recovery.

1.  **Heuristic Solver-Centered Enumerative SAT Framework:** This framework utilizes a lightweight neural network to predict a small set of $k$ "critical" variables. Subsequently, the system enumerates all $2^k$ possible assignments for these $k$ variables, generating $2^k$ simplified subproblems. These subproblems are then solved independently using conventional SAT solvers. This method achieves up to a 5x speedup on the public benchmark SAT4CryptoBench.
2.  **Neural Distinct-Based Enumerative SAT Framework:** This framework trains a discriminator to predict whether an ANF instance is satisfiable. Once a potentially correct partial assignment is identified, the framework recovers the complete key via unit propagation. This method achieves a high full-key recovery accuracy on the Simon dataset.

**Strengths:**

* The paper is well-organized and clearly written.
* The proposed frameworks offer a practical and viable path for enhancing SAT solver performance in the field of cryptanalysis. The experimental results demonstrate a significant improvement in the solving efficiency of SAT solvers for cryptographic problems.

**Weaknesses:**

* The computational cost required for generating the training dataset is not clearly specified.
* The paper does not discuss whether the value of $k$ needs to be increased as the key length grows. While Figure 3 shows that an optimal $k$ exists, the paper does not explore how this optimal $k$ scales with problem difficulty.
*  Although the Remark in Section 4.2 claims that the method does not "require knowledge of the underlying cryptographic scheme" and relies only on "symmetric clause blocks," the experiments for Framework 2 are limited to the Simon dataset. Consequently, its generalizability remains unverified.

**Questions:**

* Regarding the training of the variable selection network in the first framework: does generating the training label for each instance require a large-scale enumerative search? If so, what is the approximate computational cost of this process?
* As the key length increases, does the value of $k$ need to be adjusted? Figure 3 shows an optimal value for $k$, but the paper does not discuss how this optimal $k$ scales with problem difficulty.
* How should the term "Complementary" in the title be interpreted? The two frameworks presented—one for CNF (assisting a solver) and one for ANF (replacing a solver)—seem to be alternative solutions rather than complementary ones.
* When unit propagation stalls, the authors employ a heuristic rule ("assign 0 to the first variable in the shortest clause"). This heuristic likely has an impact on accuracy, but this impact is not evaluated in the paper.

Additionally, there are some writing issues:
* Section 3.2, line 204-205, '... we construct a derived SAT instances ...' should be '... we construct a derived SAT instance ...'
* Variable definitions are inconsistent. Section 3.3, line 242-243, "Formally, let $m$ be the length of plaintext/ciphertext and the input“, but the variable used in the subsequent text is $n$.
* Section 4.1, '... use a neural network to distinct ...' should be '... use a neural network to distinguish ...'
* Section 4.2, '... from the first two rounds clauses ...' should be '... from the first two rounds' clauses ...'
* Caption in Fig. 3, '(s/intance)' should be '(s/instance)'
* Appendix D, the phrase 'As shown in Table 4' is repeated.

---

> ### Author Response · Authors · 2025-11-21
>
> Thank you for your thoughtful feedback and for engaging with the key contributions of our work. We welcome any additional questions, and we will incorporate all necessary revisions.
>
> **W1&Q1 The computational cost required for generating the training dataset is not clearly specified. Regarding the training of the variable selection network in the first framework: does generating the training label for each instance require a large-scale enumerative search? If so, what is the approximate computational cost of this process?**
>
> ---
>
> **A.W1&Q1** Thank you for your comment. We would like to clarify that the enumeration time scales as **$32\times 2^k \times T_k$, where $T_k$ is the solving time of a derived instance given a specific assignment to $k$ variables**. The generation process is as follows: we select candidate top-k key variables based on the findings in [1]. As shown in [1], for cryptographic SAT instances, prioritizing the enumeration of variables that reduce second-order terms in the round function to first-order can significantly decrease solving time. For example, in Simon, prioritizing the reduction of the quadratic term $l_{i+1 \mod n}\cdot l_{i+8 \mod n}$ in the clause:
>
> $u_i + l_{i+1 \mod n}\cdot l_{i+8 \mod n}  +  l_{i+2 \mod n} + r_i + k_i = 0$
>
> to a linear term—by enumerating either $l_{i+1 \mod n}$ or $l_{i+8 \mod n}$ first —can substantially reduce solving time. Due to the symmetry of the round function, the variables $\{l_{i+1 \mod n}\}$ form a cyclic sequence of length 32, so the optimal strategy is to enumerate a consecutive segment of this cycle. This gives 32 candidate variable selection strategies, and our neural-assisted variable selection essentially solves this as a 32-class classification problem. Therefore, the total enumeration time scales as $32\times 2^k \times T_k$. Furthermore, through parallelization, the factor of 32 can be effectively amortized, **aligning the data-label generation cost with the testing time**.
>
> [1] Trimoska, Monika, Sorina Ionica, and Gilles Dequen. "Parity (XOR) reasoning for the index calculus attack." International Conference on Principles and Practice of Constraint Programming. Cham: Springer International Publishing, 2020.

---

> ### Author Response · Authors · 2025-11-21
>
> **W2&Q2 The paper does not discuss whether the value of $k$ needs to be increased as the key length grows. While Figure 3 shows that an optimal $k$ exists, the paper does not explore how this optimal $k$ scales with problem difficulty.**
>
> ---
>
> **A.W2&Q2** Thank you for your comment. **The value of $k$ does not need to grow with key length or block size**.
>
> As $k$ increases, the solving time of individual derived instances quickly reduces to the millisecond scale, making the bottleneck shift from $T_k$ to the $2^k$ enumerations. Thus, even for large block sizes, $k$ typically varies within the range of 4–12.
>
> In practice, existing solvers already struggle with 12-round, 32-bit Simon instances (maximum solving times exceeding 5000 s). Increasing the block size to 48 or 64 would make large-scale evaluation infeasible, while reducing the number of rounds significantly decreases problem difficulty (e.g., for 32-bit blocks, 11 rounds reduce solving time to ~1/10 of 12 rounds, and 10 rounds only require a few seconds). Therefore, we use the 12-round, 32-bit Simon dataset and Cipher-12 dataset, demonstrating that our approach remains practical beyond small keys.

---

> ### Author Response · Authors · 2025-11-21
>
> **W3 Although the Remark in Section 4.2 claims that the method does not "require knowledge of the underlying cryptographic scheme" and relies only on "symmetric clause blocks," the experiments for Framework 2 are limited to the Simon dataset. Consequently, its generalizability remains unverified.**
>
> ---
>
> **A.W3** Thank you for your comment. Following your suggestion, we have added experimental results evaluating our framework on cryptographic SAT instances derived from a 1-round simple AES-128 setting without MixColumns, which is structurally very different from the Simon dataset.
>
> In this simplified AES-128 model, the symmetric clause blocks arise from byte-wise operations. Once we guess and enumerate a subset of byte values, these symmetries are broken, enabling the model to distinguish among different key-dependent structures. In this dataset, we first enumerate the value of the 13th key byte, using an assignment length of 8 bits, and construct the derived datasets(K10,K6,K7) by distinguishing the induced assignments of the 10th, 6th, and 7th key bytes, which cannot be directly inferred.
>
> The results are summarized below. Notably, the original 1-round AES-128 SAT instance is too large to process without substituting multiple key values; its CNF/ANF graph cannot fit into memory under normal batch sizes (batch size 1–2 at most). Therefore, the table reports only the results of our framework at different stages of the solving process:
>
> | Solver       | K10 | K6 | K7 |
> | ------------ | ------------- | ------------- | ------------- |
> | NDESAT(ours) | 92.3%         | 94.5%         | 93.0% |
>
> These results confirm the generalizability of our method to cryptographic SAT instances beyond Simon.
>
> Additionally, we provide a further reflection: although the process requires enumerating only 8 bits at the beginning, generating each dataset for the target key byte still requires a factor of $2^8$ in data construction time. We believe this leaves room for further optimization in practice.

---

> ### Author Response · Authors · 2025-11-21
>
> **Q3 How should the term "Complementary" in the title be interpreted? The two frameworks presented—one for CNF (assisting a solver) and one for ANF (replacing a solver)—seem to be alternative solutions rather than complementary ones.**
>
> ---
>
> **A.Q3** Thank you for your comment. We refer to the two frameworks as complementary because they address different aspects of the challenges posed by cryptographic SAT solving:
>
> - **Neural-discriminator–based methods** – fast but not fully accurate
>
> - **Heuristic-solver–based methods** – fully accurate but slow
>
> Together, they provide complementary strengths in addressing the trade-offs between efficiency and accuracy.
>
> ---
>
> **Q4 When unit propagation stalls, the authors employ a heuristic rule ("assign 0 to the first variable in the shortest clause"). This heuristic likely has an impact on accuracy, but this impact is not evaluated in the paper.**
>
> ---
>
> **A.Q4** Thank you for your comment. We show the accuracy under the setting where a derived instance is treated as unsolved whenever unit propagation stalls as follows. **
>
> | Solver  | Simon-3-16-32| Simon-3-32-64 | Simon-6-16-32 | Simon-6-32-64|
> | -------- | --------- | ----- | ------ | ------ |
> |  NDESAT(ours)  | 98.4% |  97.3% |  83.0%   | 79.5% |
>
> As shown above, this does not affect the key-recovery accuracy. In essence, this heuristic rule is equivalent to assigning a random value and carries no special significance; its sole purpose is to allow the overall solving pipeline to proceed smoothly even when the neural discriminator makes an incorrect prediction.
>
> ---
>
> **About typos**
>
> ---
>
> Thank you for your careful comments. We will revise the paper accordingly and incorporate all necessary updates.

---

### Official Review · Reviewer_dnEM · 2025-10-30

**Soundness:** 3
**Presentation:** 4
**Contribution:** 2
**Rating:** 4
**Confidence:** 2

**Summary:**

The paper explores applying neural guidance to SAT-based cryptographic key recovery, noting that conventional CDCL solvers struggle with these instances due to symmetry and redundancy. To address this, the authors propose two complementary enumerative frameworks: Heuristic Solver-Centered Enumeration (HSESAT) uses a neural model to predict the top-k key variables for enumeration, with fixed subproblems solved by a classical SAT solver. Neural Distinct-Based Enumeration (NDESAT), for ANF representations, learns to distinguish structurally different instances to predict the correct key, using unit propagation to complete assignments. Experiments on SAT4CryptoBench show HSESAT achieves substantial speedups over Kissat/CryptoMiniSat, and NDESAT demonstrates high key-recovery success rates.

**Strengths:**

1. Two frameworks tailored for different representations (CNF vs. ANF) with complementary goals (solver acceleration vs. key recovery)
2. Strong empirical results compared with NeuroSAT and CryptoANFNet baselines

**Weaknesses:**

1. HSESAT requires enumeration preprocessing to determine which k-variable combinations yield the fastest solving. For each training instance, this implies evaluating $C(n,k)* 2^k$ solver runs, which can be extremely expensive. The paper does not provide any runtime or resource analysis of this offline phase, leaving unclear whether the approach is practical beyond small keys.
2. The paper never reports the top-k prediction accuracy of the neural model or its correlation with actual solver speedups. Without these statistics, it is difficult to evaluate whether the network genuinely learns meaningful structural signals or if the gains mainly come from enumeration randomness.
3. The paper lacks a deeper insight explaining *why* these integrations succeed where prior neural-SAT attempts fail (e.g., what structural cues the model captures, what distinguishes “hard” and “easy” key variables). The stated insight—that “placing the neural model outside the solver loop reduces overhead”—is useful but not particularly surprising.

**Questions:**

1. Both frameworks depend on a manually chosen k. Is there a way to adaptively adjust k during inference or training?
2. The paper reports average per-instance solving time for the enumerated subproblems, but not the overall runtime or resource usage. Could the authors clarify whether solving 2^k derived instances actually reduces the total computational cost under fixed hardware settings, or whether the observed speedup mainly arises from parallel execution?

---

> ### Author Response · Authors · 2025-11-21
>
> Thank you for your thoughtful feedback and for engaging with the key contributions of our work. We welcome any additional questions, and we will incorporate all necessary revisions.
>
> ---
>
> **Q2 The paper reports average per-instance solving time for the enumerated subproblems, but not the overall runtime or resource usage. Could the authors clarify whether solving 2^k derived instances actually reduces the total computational cost under fixed hardware settings, or whether the observed speedup mainly arises from parallel execution?**
>
> **A.Q2** Thanks for this comment. This is an important misunderstanding. In Table 1, the time for base refers to **the solving time of the original instance** on the baseline solver, while the times for random and HSESAT represent **the total solving time of the $2^k$ derived instances from the original instance**, rather than the solving time of a single subproblem.

---

> ### Author Response · Authors · 2025-11-21
>
> **W1 HSESAT requires enumeration preprocessing to determine which k-variable combinations yield the fastest solving. For each training instance, this implies evaluating $C(n,k)\times 2^k$ solver runs, which can be extremely expensive. The paper does not provide any runtime or resource analysis of this offline phase, leaving unclear whether the approach is practical beyond small keys.**
>
> ---
>
> **A.W1** Thank you for this comment. We would like to clarify that the enumeration time scales as **$32\times 2^k \times T_k$, where $T_k$ is the solving time of a derived instance given a specific assignment to $k$ variables**. The generation process is as follows: we select candidate top-k key variables based on the findings in [1]. As mentioned in [1], for cryptographic SAT instances, prioritizing the enumeration of variables that reduce second-order terms in the round function to first-order can significantly decrease solving time. For example, in Simon, prioritizing the reduction of the quadratic term $l_{i+1 \mod n}\cdot l_{i+8 \mod n}$ in the clause:
>
> $u_i + l_{i+1 \mod n}\cdot l_{i+8 \mod n}  +  l_{i+2 \mod n} + r_i + k_i = 0$
>
> to a linear term—by enumerating either $l_{i+1 \mod n}$ or $l_{i+8 \mod n}$ first —can substantially reduce solving time. Due to the symmetry of the round function, the variables $\{l_{i+1 \mod n}\}$ form a cyclic sequence of length 32, so the optimal strategy is to enumerate a consecutive segment of this cycle. **This gives 32 candidate variable selection strategies, and our neural-assisted variable selection essentially solves this as a 32-class classification problem.** Therefore, the total enumeration time scales as $32\times 2^k \times T_k$. Furthermore, through parallelization, the factor of 32 can be effectively amortized, **aligning the data-label generation cost with the testing time**.
>
> Additionally, **the value of $k$ does not need to grow with key length or block size**. As $k$ increases, the solving time of individual derived instances quickly reduces to the millisecond scale, making the bottleneck shift from $T_k$ to the $2^k$ enumerations. Thus, even for large block sizes, $k$ typically varies within the range of 4–12. In practice, existing solvers already struggle with 12-round, 32-bit Simon instances (maximum solving times exceeding 5000 s). Increasing the block size to 48 or 64 would make large-scale evaluation infeasible, while reducing the number of rounds significantly decreases problem difficulty (e.g., for 32-bit blocks, 11 rounds reduce solving time to ~1/10 of 12 rounds, and 10 rounds only require a few seconds). Therefore, we use the 12-round, 32-bit Simon dataset and Cipher-12 dataset, demonstrating that our approach remains practical beyond small keys.
>
> [1] Trimoska, Monika, Sorina Ionica, and Gilles Dequen. "Parity (XOR) reasoning for the index calculus attack." International Conference on Principles and Practice of Constraint Programming. Cham: Springer International Publishing, 2020.

---

> ### Author Response · Authors · 2025-11-21
>
> **W2 The paper never reports the top-k prediction accuracy of the neural model or its correlation with actual solver speedups. Without these statistics, it is difficult to evaluate whether the network genuinely learns meaningful structural signals or if the gains mainly come from enumeration randomness.**
>
> ---
>
> **A.W2** Thank you for your comment. As discussed in our response to W1, we select the candidate top-k key variables based on the approach in [1]. Essentially, this can be viewed as a cyclic selection problem over the round-function clause blocks, equivalent to a 32-class classification problem corresponding to the choice of the starting point of the selection sequence. We report the prediction accuracy as follows:
>
> | Mehtod  | Simon-12-32-64| Cipher-12 |
> | -------- | --------- | ----- |
> |  Neural-assisted variable selection  | 74.1% |  73.4% |
>
> Additionally, in the original paper, we have already provided results under random selection of these 32 classes, as shown in the ***random*** line in Table 1. While random selection can achieve some improvement in solving time for certain solvers, such as Kissat, it leads to negative or negligible effects on others, including MapleSAT and CryptoMiniSat. In contrast, our enumeration framework guided by neural-assisted selection is able to adapt and generalize across different solvers, consistently delivering larger performance gains. Therefore, we conclude that the observed improvements are not due to enumeration randomness.
>
> [1] Trimoska, Monika, Sorina Ionica, and Gilles Dequen. "Parity (XOR) reasoning for the index calculus attack." International Conference on Principles and Practice of Constraint Programming. Cham: Springer International Publishing, 2020.

---

> ### Author Response · Authors · 2025-11-21
>
> **W3 The paper lacks a deeper insight explaining why these integrations succeed where prior neural-SAT attempts fail (e.g., what structural cues the model captures, what distinguishes “hard” and “easy” key variables). The stated insight—that “placing the neural model outside the solver loop reduces overhead”—is useful but not particularly surprising.**
>
> ---
>
> **A.W3**  Thank you for your comment. We will clarify the insight explaining why our integrations succeed.
>
> Although there are several ML-based heuritic solvers, cryptographic SAT instances are notoriously difficult for them, as explained in Section 3.1:
>
> **(1) The search space of round-reduced block ciphers exhibits strong symmetries, undermining standard activity heuristics.**
>
> **(2) Solver branching decisions frequently oscillate on low-information intermediate variables, leading to exponential blow-ups.**
>
> Furthermore, repeated solver–ML interactions often incur significant time overhead. Thus, existing ML-based solvers still struggle with efficiency.
>
> The core purpose of our method is to address **the challenge faced by traditional or ML-based heuristic solvers: namely, the implicit learning of variable dependencies (decision priorities) in cryptographic SAT formulas**. Cryptographic SAT instances exhibit strong structural features induced by the round function. In this setting, traditional heuristics or ML-based modules that read the entire CNF for guidance are less direct and effective than our approach, which **explicitly guides variable selection based on plaintext-ciphertext pairs and the round-function structure**.
>
> We model this task as a 32-class classification problem, with the plaintext-ciphertext pairs as input, as described in A.W1. The differences among these 32 options arise because the constant terms in the first two rounds of the round-function clauses vary across different plaintext-ciphertext pairs, naturally making some selections of {$l_{i+1 \mod n}$} more efficient than others.

---

> ### Author Response · Authors · 2025-11-21
>
> **Q1 Both frameworks depend on a manually chosen k. Is there a way to adaptively adjust k during inference or training?**
>
> ---
>
> **A.Q1** Thank you for your comment. The treatment of $k$ differs across our two frameworks.
>
> **For the Heuristic solver-centered framework, $k$ is a hyperparameter.** We select its value using an independent validation set. Since there is an explicit trade-off between $k$ and the average solving time of the derived instances—and the feasible values of $k$ are discrete—we can tune it using simple hyperparameter optimization methods (e.g., the two optimization tools provided in SAT4CryptoBench). In practice, however, we adopt an even simpler strategy: we evaluate the average solving time on an independent validation set for small range of $k$ (e.g. $k \in [4,12]$) and choose the value that yields the lowest time. Importantly, this means the $k$ reported in Table 1 is not chosen by manually inspecting the final test-set results, but is determined beforehand using the validation set.
>
> **For the Neural distinct-based framework, the choice of $k$ comes directly from analyzing the structure of the round-function clauses.** Specifically, we examine how substituting values can break the inherent symmetry in these clauses, enabling the neural network to discriminate effectively. Thus, in this setting, $k$ is determined by analyzing the SAT instances in the dataset.
>
> **If the concern is whether $k$ can be selected adaptively during solving**, we believe that techniques inspired by multi-armed bandit–based adaptive restart strategies may in principle be applied. However, we remain cautious: evaluating the quality of different $k$ values is substantially more expensive than assessing restart strategies based on past solving history. Integrating such a mechanism would likely introduce considerable overhead. In contrast, our current approach is more practical and efficient: we adaptively select $k$ once using a structurally similar validation set, and then evaluate on the full test set. In this way, $k$ works as a standard hyperparameter and does not incur any additional cost during testing.

---

### Official Review · Reviewer_5uif · 2025-10-31

**Soundness:** 3
**Presentation:** 4
**Contribution:** 2
**Rating:** 4
**Confidence:** 2

**Summary:**

This paper proposes two neural-guided enumerative SAT frameworks for cryptographic key recovery. The first predicts a small set of influential key bits and enumerates their assignments before invoking a standard SAT solver, yielding up to 5× speed-ups on SAT4CryptoBench. The second uses a neural discriminator to identify the correct assignment among 2^k ANF-derived instances and then performs unit-propagation to recover full keys, achieving ~80% full-key recovery on Simon-6-32-64 — substantially outperforming prior neural approaches.

**Strengths:**

1. This paper is well-written and easy to follow.

2. Strong empirical results: up to 5× faster solving and over 79.5% key-recovery accuracy on real crypto benchmarks.

3. General framework: solver-agnostic and adaptable to different ciphers/representations (CNF/ANF).

**Weaknesses:**

1. I recommend providing a more comprehensive and deeper comparative analysis of the two proposed methods. In particular, it would be valuable to clarify under what circumstances each method is preferred, and to present direct comparisons on the same benchmark settings to highlight their respective strengths and limitations. Additionally, please discuss whether the two methods can be combined into a unified pipeline; if not, clarifying the practical or conceptual barriers to doing so would strengthen the contribution.

2. Regarding HSESAT, could the authors clarify the fundamental reason why it achieves faster performance than existing solvers? The method appears to rely on a strong assumption that a set of “top-k” key variables exists and can be identified reliably. How robust is HSESAT when this assumption does not hold — for example, when variable scores are relatively uniform and no clear top-k variables emerge? Additionally, can HSESAT generalize beyond SAT4CryptoBench? In particular, would it still provide benefits on broader SAT benchmarks such as SATLIB (https://www.cs.ubc.ca/~hoos/SATLIB/), where cryptographic structure and key-bit symmetry are absent? A discussion or experiment on general SAT datasets would help clarify the scope and applicability of the method.

3. Figure 1 suggests HSESAT outputs the key, but the evaluation only reports solver runtime in Table 1. It would strengthen clarity to either (a) report key-recovery accuracy for HSESAT, or (b) explicitly state that HSESAT uses solver correctness as a proxy for key recovery.

4. I recommend to add failure analysis for Table 2. Why the remaining keys fail in Table 2. The relevant section simply states high recovery accuracy and mentions the method’s limitations compared to heuristic solvers, without analyzing failure causes.

5 (Minor) It would be beneficial to include a brief discussion of SAT hardware accelerators in the background section and clarify whether such hardware efforts could be complementary to the proposed approaches. This would provide a more comprehensive view of the broader SAT-solving landscape.

[1] S. Su et al., "A Stochastic Analog Boolean Satisfiability Solver," in IEEE Journal of Solid-State Circuits

[2] Wu, Zihan, et al. "37.5 SKADI: A 28nm Complete K-SAT Solver Featuring Dual-Path SRAM-Based Macro and Incremental Update with 100% Solvability." 2025 IEEE International Solid-State Circuits Conference (ISSCC). Vol. 68. IEEE, 2025.

**Questions:**

Please see the weakness above.

---

> ### Author Response · Authors · 2025-11-21
>
> Thank you for your thoughtful feedback and for engaging with the key contributions of our work. We welcome any additional questions, and we will incorporate all necessary revisions.

---

> ### Author Response · Authors · 2025-11-21
>
> **W1 I recommend providing a more comprehensive and deeper comparative analysis of the two proposed methods. ..**
>
> ---
>
> **A.W1** Thanks for the comment. Based on Section 2, 3.1, and 4.1, we will further clarify this issue.
>
> Our two frameworks are motivated by the limitations of existing approaches. Current methods fall into two main categories:
>
> **1. Neural-discriminator–based methods**
>
> As summarized in ***Neural SAT-based Cryptanalysis in Section 2***, these methods recover the key bit-by-bit. For the $i$-th key bit, they assign the corresponding SAT variable to 0/1 to obtain a SAT–UNSAT derived-instance pair, then train a neural network to predict satisfiability. The predicted satisfiable assignment is treated as the recovered bit. Repeating this for all bits essentially turns key recovery into an end-to-end classification task.
>
> However, **as discussed in Section 4.1**, although these methods are fast, their per-bit SAT/UNSAT classification accuracy remains low (around 60\%). Consequently, full key recovery becomes nearly impossible—for a 16-bit key, the expected accuracy is only $0.6^{16}<0.02\%$.
>
> A related line of work, summarized in ***Differential Cryptanalysis with Neural Networks in Section 2***, uses neural networks to classify ciphertext pairs generated under certain plaintext/key differences, combined with enumeration.
>
> While networks accelerate filtering, the enumeration cost remains large (close to $2^{31}$), and these methods are unsuitable for single plaintext–ciphertext SAT-based key-recovery scenarios. They also require very large datasets (>$10^7$) and their key-recovery accuracy is far below that of complete SAT solvers.
>
> Compared with these methods, our neural distinct-based framework improves full-key recovery by generating distinguishable derived instances and combining neural distinction with unit propagation.
>
> Enumeration during derived-instance construction breaks the inherent symmetry of SAT instances in cryptographic settings, improving discriminative accuracy. Moreover, by assigning $k$ variables, unit propagation can fully solve each derived instance, avoiding multiplicative error accumulation from multiple predictions. This achieves both comparable speed and higher recovery accuracy—representing a substantive advancement over prior neural-discriminator-based methods.
>
> **2. Heuristic-solver-based methods**
>
> As summarized in ***ML-based heuristic SAT Solvers in Section 2***, ML-based SAT solvers optimize parts of the solving process inside traditional SAT solvers. Although there are several ML-based heuristic solvers, cryptographic SAT instances are notoriously difficult for them, **as explained in Section 3.1**:
>
> **(1) The search space of round-reduced block ciphers exhibits strong symmetries, undermining standard activity heuristics.**
>
> **(2) Solver branching decisions frequently oscillate on low-information intermediate variables, leading to exponential blow-ups.**
>
> Furthermore, repeated solver–ML interactions often incur large time overhead. Thus, existing ML-based solvers still struggle with efficiency.
>
> Our heuristic solver-centered framework integrates a single neural-guided variable-selection step with classical heuristics. The network selects $k$ informative variables, which are explicitly enumerated before invoking the solver. This reduces the effective search space and dramatically lowers solving time, while preserving the generalization and completeness of the underlying solver.
>
> **About Baselines and Datasets**
>
> The two frameworks originate from two complementary methodological lines:
>
> - **Neural-discriminator–based methods** – fast but not fully accurate
>
> - **Heuristic-solver–based methods** – fully accurate but slow
>
> Thus, the baselines and evaluation metrics must match the characteristics of each method.
>
> - **Heuristic solver-centered framework**:
> This framework is evaluated on the hardest key-recovery instances in SAT4CryptoBench. Baselines include the heuristic solvers (MapleSAT, Kissat, CryptoMiniSat).
> Here, in Table 1, all results achieve 100% accuracy. The time for *base* refers to ***the solving time of the original instance on baseline solver***, while the times for *random* and *HSESAT* represent ***the total solving time of the $2^k$ derived instances from the original instance***, rather than the solving time of a single subproblem.
>
> - **Neural distinct-based framework**:
> This framework is compared with neural end-to-end solvers (NeuroSAT, CryptoANFNet). Due to GPU memory limits, evaluations are on the 3/6 rounds Simon dataset. Here, in Table 2, the accuracy refers to the accuracy of full key recovery.
>
> Now, neural-discriminator-based methods are restricted by accuracy and memory and cannot scale to the larger datasets required for heuristic solver-based baselines. Therefore, cross-category comparison is currently infeasible. We anticipate that our two frameworks will inspire further developments that eventually narrow this gap.

---

> ### Author Response · Authors · 2025-11-21
>
> **W2 Regarding HSESAT, could the authors clarify the fundamental reason why it achieves faster performance than existing solvers? The method appears to rely on a strong assumption that a set of “top-k” key variables exists and can be identified reliably. How robust is HSESAT when this assumption does not hold — for example, when variable scores are relatively uniform and no clear top-k variables emerge? Additionally, can HSESAT generalize beyond SAT4CryptoBench? In particular, would it still provide benefits on broader SAT benchmarks such as SATLIB (https://www.cs.ubc.ca/~hoos/SATLIB/), where cryptographic structure and key-bit symmetry are absent? A discussion or experiment on general SAT datasets would help clarify the scope and applicability of the method.**
>
> ---
>
> **A.W2** Thank you for your comment. **As discussed in  A.W1**, HSESAT selects
> $k$ informative variables (the key variables of the cipher in our setting), which are explicitly enumerated before invoking the solver. This effectively reduces the search space. The core rationale behind this approach lies in the structural symmetry inherent in cryptographic SAT instances, and the advantage of enumerating from key variables to break substructures while allowing the solver to avoid enumerating intermediate variables that carry little information. This is a property intrinsic to the cryptographic SAT instances themselves, rather than an additional assumption. As shown in Table 1, even for the Cipher datasets where the encryption details are unknown, HSESAT performs effectively.
>
> For SATLIB, we tested the three selected solvers on the largest and most challenging datasets as described in the SATLIB benchmark. The results are as follows (s/instance):
>
> | Solver  | Uniform Random-3-SAT(uf250-1065)| Uniform Random-3-SAT(uuf250-1065) | Random-3-SAT(CBS_k3_n100_m449_b90) | "Flat" Graph Colouring(flat200-479)| "Morphed" Graph Colouring(sw100-8-lp8-c5) |
> | -------- | --------- | ----- | ------ | ------ | ------ |
> |  Kissat  | 0.641 |  3.650 |  0.007   | 0.011 | 0.005 |
> |  MapleSAT  | 1.300 |  3.640 |  0.007   | 0.015 | 0.009 |
> |  CryptoMiniSat  | 2.416 |  13.560 |  0.070   | 0.070 | 0.060 |
>
> It is clear that the SATLIB datasets are far simpler compared to cryptographic SAT instances. All three baseline solvers already achieve extremely fast solving times (which can be viewed as the special case of $k=0$). Therefore, in this work, we do not use such datasets for evaluation.
>
> Of course, if one assumes that an SAT dataset lacks structural symmetry, selecting $k$ variables for enumeration would merely serve as a heuristic. Applied to datasets without this property, enumerating $k$ variables may even increase solving time.
>
> Importantly, in the setting of key recovery, structural symmetry is always present. Therefore, this method is generally applicable to SAT instances arising from cryptography. Discussions outside this setting are not the focus of this work. Nonetheless, for SAT datasets that do exhibit structural symmetry, HSESAT can still be effectively applied. Core variables in structurally similar subclauses and the constants in them can be used to select the most informative $k$ variables for enumeration.

---

> ### Author Response · Authors · 2025-11-21
>
> **W3 Figure 1 suggests HSESAT outputs the key, but the evaluation only reports solver runtime in Table 1. It would strengthen clarity to either (a) report key-recovery accuracy for HSESAT, or (b) explicitly state that HSESAT uses solver correctness as a proxy for key recovery.**
>
> ---
>
> **A.W3** Thank you for your comment. **As discussed in A.W1**, HSESAT is built upon classical SAT solvers and correctly solves all original instances in the Table 1 dataset—that is, it successfully recovers the full key for every instance. Its baselines are complete SAT solvers, and the comparison metric is solving time. All results reported in Table 1 achieve 100% solving accuracy.
>
> ---
>
> **W4 I recommend to add failure analysis for Table 2. Why the remaining keys fail in Table 2. The relevant section simply states high recovery accuracy and mentions the method’s limitations compared to heuristic solvers, without analyzing failure causes.**
>
> ---
>
> **A.W4** Thank you for your comment. We provide a concise clarification and failure analysis of the solvers in Table 2.
>
> **NeuroSAT & CryptoANFNet**: NDESAT is an end-to-end neural discriminator, compared against NeuroSAT and CryptoANFNet. These baselines recover the key by predicting each bit independently: for every key bit, they generate two nearly identical SAT instances (bit = 0/1) and let the network discriminate between them. However, **assigning a single bit changes only 1–3 clauses, meaning the two CNF/ANF graphs differ by only a few edges.** As a result, single-bit discrimination is extremely difficult (NeuroSAT fails; CryptoANFNet achieves only ~60%), and **errors multiply across 16–32 bits, leading to low final key-recovery accuracy**.
>
> NDESAT overcomes this by analyzing the round-function structure to identify an assignment length $k$ that meaningfully breaks symmetry. We then generate $2^k$ structurally distinct derived instances: incorrect assignments typically fail to break the symmetry, while the correct one does, allowing reliable discrimination. During testing, we evaluate all $2^k$ derived instances, choose the most likely assignment, and finalize the solution via unit propagation—avoiding any multiplicative error accumulation.
>
> **NDESAT**: Failures of our NDESAT occur only when the neural network misclassifies derived instances in cases where **substituting values into only the first three rounds does not introduce sufficient structural difference**. Extending substitution to more rounds would eliminate these cases but would make the process closer to heuristic SAT solving and lose the computational advantages of neural approaches. Limiting substitution to three rounds strikes a practical balance between accuracy and efficiency.
>
> ---
>
> **W5 It would be beneficial to include a brief discussion of SAT hardware accelerators in the background section and clarify whether such hardware efforts could be complementary to the proposed approaches. This would provide a more comprehensive view of the broader SAT-solving landscape.**
>
> **A.W5** Thank you for the helpful suggestion. Although SAT hardware accelerators are not the primary focus of this work, they indeed represent a complementary line of research. Recent advances—including stochastic analog SAT accelerators [1] and complete on-chip K-SAT solvers such as SKADI with dual-path SRAM-based architectures [2]—demonstrate that specialized hardware can significantly speed up clause evaluation, propagation, and incremental updates.
>
> While these works address performance at the hardware level, our proposed methods focus on algorithmic and solver-level improvements. The two directions are orthogonal and potentially synergistic: techniques such as derived-instance construction or neural-guided variable selection could in principle be integrated with hardware-accelerated SAT engines to further reduce solving time.
>
> We will include this discussion in our paper.
>
> [1] S. Su et al., "A Stochastic Analog Boolean Satisfiability Solver," in IEEE Journal of Solid-State Circuits
>
> [2] Wu, Zihan, et al. "37.5 SKADI: A 28nm Complete K-SAT Solver Featuring Dual-Path SRAM-Based Macro and Incremental Update with 100% Solvability." 2025 IEEE International Solid-State Circuits Conference (ISSCC). Vol. 68. IEEE, 2025.

---

### Author Response · Authors · 2025-11-28

Dear Reviewers,

We hope this message finds you well.

We have submitted our rebuttal responses to address all suggestions and clarify any remaining ambiguities or misunderstandings.

With the rebuttal deadline in four days, we kindly ask whether our responses have resolved your main concerns or if any further issues require attention.

Thank you very much for your time and valuable feedback.

Best regards,

The Authors

---

### Author Response · Authors · 2025-12-03
**General Response**

We sincerely appreciate the reviewers’ time, feedback, and constructive suggestions. Overall, our work was deemed “well-organized/written” (`5uif, dnEM, xeJ3`) with an “excellent”  presentation and easy to follow (`5uif, dnEM`). The results were described as “strong empirical,” (`5uif, dnEM`) and “significant improvement” (`xeJ3`).

---

Our work introduces two complementary frameworks for SAT-based cryptanalysis:

**Neural Distinct-Based Framework (NDESAT)**

- Targets neural discriminator–style methods and focuses on generating structurally distinct derived instances to improve neural discriminability.

- Avoids multiplicative prediction errors during full key recovery.

- Achieves higher full-key recovery accuracy than prior neural end-to-end solvers.

**Heuristic Solver-Centered Framework (HSESAT)**

- Targets classical and ML-enhanced SAT solvers, focusing on reducing the effective search space through neural-assisted variable selection followed by enumeration.

- Retains solver completeness while significantly reducing solving time on cryptographic instances.

All in all, the frameworks tackle two opposite ends of the cryptographic SAT-solving spectrum:

| Method class | Advantages | Weaknesses | Our improvements |
| ------------ | ---------- | ---------- | ----------------
Neural discriminator–based approaches|	Fast inference	|Cannot scale; accuracy too low to recover full keys|	Break symmetry through derived-instance construction; k-bit substitution eliminates per-bit multiplicative errors|
|Heuristic SAT solvers|	Always correct for completely solving|	Slow; struggle with cipher symmetry; activity heuristics misled	| Select k informative variables to enumerate; drastically reduces search depth|

---

**Across all reviews, we have clarified these major misunderstandings:**

1. **About the choice of k(`dnEM,xeJ3`):**

For HSESAT, $k$ is a standard hyperparameter tuned on a validation set.
For NDESAT, $k$ is determined by structural properties of the cipher, specifically the granularity at which symmetric clause blocks arise. Importantly,
**k$ does not scale with the key size**, and this distinction is now made explicit.

**(See A.Q1 in response to dnEM & A.W2&Q2 in response to xeJ3)**

2. **About training cost and enumeration(`dnEM,xeJ3`):**

Several reviewers raised concerns about the cost of generating training labels for variable selection. We clarify that, leveraging cyclic symmetries in the round functions, the enumeration reduces to a **32-class classification problem**, making the offline cost manageable (**$32\times 2^k \times T_k$, where $T_k$ is the solving time of a derived instance given a specific assignment to $k$ variables**) and **aligned with the testing-time scale**.

**(See A.W1 in response to dnEM and A.W1&Q1 in response to xeJ3)**

4. **About generalizability of NDESAT(`xeJ3`):**

Reviewer xeJ3 questioned whether NDESAT generalizes beyond Simon. We evaluated it on **1-round AES-128**—structurally distinct from Simon, with byte-wise symmetric clause blocks. After enumerating the **13th key byte (8 bits)**, these symmetries are broken, allowing the model to distinguish induced assignments of the 10th, 6th, and 7th key bytes (K10, K6, K7), which cannot be inferred directly.
The results demonstrate that NDESAT generalizes effectively to cryptographic SAT instances beyond Simon. While constructing each derived dataset requires a $ 2^8$-factor increase in preprocessing, this remains practical and leaves room for future optimization.

**(See A.W3 in response to xeJ3)**

5. **About baseline selection, dataset scope and evalution(`5uif,dnEM`):**

The two frameworks correspond to two distinct methodological lines, each requiring different baselines:

- **Heuristic solver-centered framework**:
This framework is evaluated on the hardest key-recovery instances in SAT4CryptoBench. Baselines include the heuristic solvers (MapleSAT, Kissat, CryptoMiniSat).
In Table 1, all results achieve 100% accuracy. The time for *base* refers to ***the solving time of the original instance on baseline solver***, while the times for *random* and *HSESAT* represent ***the total solving time of the $2^k$ derived instances from the original instance***, rather than the solving time of a single subproblem.

- **Neural distinct-based framework**:
This framework is compared with neural end-to-end solvers (NeuroSAT, CryptoANFNet). Due to GPU memory limits, evaluations are on the 3/6 rounds Simon dataset. In Table 2, the accuracy refers to full key recovery.

Besides, we clarify that general SAT benchmarks (like SATLIB) lack the structural symmetries inherent to cryptographic problems and are thus unsuitable for evaluating cryptographic SAT-solving performance.

**(See A.W1-W4 in response to 5uif and A.Q2 in response to dnEM)**

---

We fully acknowledge the challenges faced by the ICLR community and the additional workload placed on the ACs. Sincere thanks again to the reviewers and ACs for their efforts.

---

### Meta-Review · Area_Chair_ffNL · 2025-12-22

**Summary:**

This paper proposes two neural-guided enumerative SAT frameworks for cryptographic key recovery, targeting the infamous difficulty of applying classical CDCL solvers and existing neural SAT methods to cryptographic instances with strong structural symmetry. The first framework, HSESAT, augments classical SAT solvers with a lightweight neural model that selects a small set of informative key variables to enumerate, reducing effective search depth while preserving solver completeness. The second, NDESAT, focuses on ANF encodings and uses a neural discriminator over structurally distinct derived instances to identify correct partial assignments, followed by unit propagation to recover full keys. Together, the frameworks aim to bridge the gap between fast but inaccurate neural approaches and exact but slow SAT solvers.

Across reviews, the paper was consistently evaluated as "marginally below the acceptance threshold", although reviewers recognized the practicality of combining neural guidance with explicit enumeration rather than embedding ML inside the solver loop. Initial concerns centered on the cost and scalability of enumeration-based training, the choice and role of the hyperparameter k, the scope of generalization beyond Simon, clarity of evaluation metrics, and whether the two frameworks are “complementary.” The authors provided a detailed rebuttal, added new experiments (including AES-based instances and SATLIB controls), clarified evaluation protocols, and addressed misunderstandings about runtime accounting and accuracy.

**Reviewer Concerns:**

Shared comments:

1. Cost and Practicality of Enumeration-Based Training. Multiple reviewers questioned whether generating training labels—especially for HSESAT’s variable-selection model—requires prohibitively expensive enumeration, potentially undermining scalability. Resolution: The authors clarified that enumeration exploits strong cipher symmetries, reducing the problem to a small fixed classification task (e.g., 32 candidate variable blocks in Simon). The offline cost scales as a small constant factor times the solving time of derived instances and can be amortized via parallelization. Importantly, k does not grow with key size, and practical values remain small (typically 4–12).

2. Choice, Role, and Scaling of the Hyperparameter k. Reviewers **repeatedly** asked how k is chosen, whether it must increase with key length, and whether it can be adapted dynamically. Resolution: The authors clarified that k plays different roles in the two frameworks: it is a tunable hyperparameter selected on a validation set for HSESAT, and a structure-driven design choice for NDESAT. Empirically, k does not scale with key length or block size, as increasing k rapidly reduces subproblem solving time while shifting the bottleneck to enumeration.

3. Generalization Beyond Simon. Several reviewers questioned whether NDESAT generalizes beyond the Simon cipher, given its reliance on structural symmetry. Resolution: The authors added new experiments on a simplified 1-round AES-128 setting, demonstrating that the method generalizes to cryptographic SAT instances with different symmetry structures (e.g., byte-wise clause blocks).

4. Evaluation Clarity and Runtime Accounting. Reviewers raised concerns about whether reported runtimes reflect total computation (including enumeration) and whether accuracy metrics were clearly defined for each framework. Resolution: The authors clarified that all reported times for HSESAT reflect total solving time across all derived instances, not per-subproblem time, and that all solver-based experiments achieve 100% accuracy by construction.

Reviewer-Specific Questions:

1. Reviewer 5uif: Requested deeper comparison between the two frameworks, clarity on when each should be used, robustness of the “top-k” assumption, failure analysis, and broader context (e.g., SAT hardware). The authors provided a clear conceptual separation between the two approaches, detailed why each addresses complementary limitations of prior work, added failure analysis for NDESAT, clarified HSESAT’s accuracy guarantees, and discussed hardware accelerators as complementary.

2. Reviewer dnEM: Focused on offline training cost, lack of reported top-k prediction accuracy, and whether speedups stem from enumeration randomness. The authors quantified training cost, reported prediction accuracy (~74%), and showed that random enumeration performs inconsistently across solvers, whereas neural-guided selection yields robust gains.

3. Reviewer xeJ3: Questioned generalizability, scaling of k, interpretation of “complementary,” and heuristic choices when unit propagation stalls. The authors added AES experiments, clarified that the two frameworks address opposite ends of the accuracy–efficiency spectrum, and showed that the fallback heuristic does not affect recovery accuracy.

**Reviewer Scores:**

The scores of all reviewers are reasonable.

---

### Decision · Program_Chairs · 2026-01-26

Reject